# Central Nervous System Tissue Regeneration after Intracerebral Hemorrhage: The Next Frontier

**DOI:** 10.3390/cells10102513

**Published:** 2021-09-23

**Authors:** Ruiyi Zhang, Mengzhou Xue, Voon Wee Yong

**Affiliations:** 1The Hotchkiss Brain Institute and the Department of Clinical Neurosciences, University of Calgary, Calgary, AB T3A 4X9, Canada; ruiyi.zhang@ucalgary.ca; 2The Second Affiliated Hospital of Zhengzhou University, Zhengzhou 450014, China

**Keywords:** intracerebral hemorrhage, tissue regeneration, neurogenesis, remyelination, angiogenesis, neuroinflammation, drug therapy, stem cells, rehabilitation, biomaterial

## Abstract

Despite marked advances in surgical techniques and understanding of secondary brain injury mechanisms, the prognosis of intracerebral hemorrhage (ICH) remains devastating. Harnessing and promoting the regenerative potential of the central nervous system may improve the outcomes of patients with hemorrhagic stroke, but approaches are still in their infancy. In this review, we discuss the regenerative phenomena occurring in animal models and human ICH, provide results related to cellular and molecular mechanisms of the repair process including by microglia, and review potential methods to promote tissue regeneration in ICH. We aim to stimulate research involving tissue restoration after ICH.

## 1. Introduction

Intracerebral hemorrhage (ICH) accounts for 12–20% of all types of stroke with over 2 million individuals worldwide being afflicted annually [1]. ICH has catastrophic outcomes with up to 50% mortality and 70% disability among the survivors a year after onset [2]. A study of global disease burden shows that ICH cases have increased by 47% over the past 20 years, occurring mostly in low-income and middle-income countries [3]. This life-threatening stroke subtype can be induced by a variety of causes, including hypertension, cerebral amyloid angiopathy, trauma, vascular malformations, tumors, premature birth, and with certain drugs [4]. Even though the development of minimally invasive surgery targeting the primary injury has alleviated neurological deficits or reduced mortality [5,6,7], the prognosis of ICH remains unsatisfactory. Accordingly, scholars have turned their focus on mitigating ICH-induced inflammation and consequent secondary brain injury with significant promise preclinically and some results being clinically translated [8].

The restorative capability of the central nervous system (CNS) after ICH has received little attention, even though it is clear that the brain has capacity for repair after injury [9,10,11]. The dynamic changes of myelin (de- and remyelination) can be found in brains of patients with multiple sclerosis and Alzheimer’s disease [12,13]; a novel transgenic reporter mouse line shows proof of myelin renewal in normal homeostasis [14]. Enhanced neural regenerative processes including neurogenesis, angiogenesis, oligodendrogenesis, and axonal regeneration have been observed in divergent CNS pathologies [15,16,17]. Moreover, it is commonly observed that patients with ICH gradually recover some neurofunctional deficits several months after the stroke [18]. Such a phenomenon suggests that reorganization or regeneration of neural elements occurs after ICH, giving optimism that tissue recovery in ICH may be promoted to improve its prognosis. To achieve this goal, the regenerative events in ICH at the cellular and molecular level, and the mechanisms thereof, must be better understood. In this review, we have collated available evidence that regenerative processes occur after ICH in animal models, and we describe the sparse information currently available for patients with ICH. We describe these changes, evaluate the enabling mechanisms, and discuss potential methods to facilitate CNS repair in ICH.

## 2. Regenerative Processes Occur after ICH in Animal Models

Evidence from animal models supports the occurrence of brain regenerative processes after ICH and which are detectable by 72 h in multiple forms [19]. For neurogenesis, enhanced proliferation of neural progenitor cells (NPCs) in the subventricular zone (SVZ) and striatum is observed 14 days after collagenase-induced ICH in the striatum near the internal capsule. Both bromodeoxyuridine (BrdU, an index of cell proliferation) or doublecortin (DCX, a marker of neuroblasts) single-positive cells are dramatically increased in the ipsilateral and contralateral hemisphere of ICH compared to the sham group, with some being double positive [20,21]. Another group also demonstrates neurogenesis after collagenase-induced ICH through the robust increase in Ki67^+^/DCX^+^ and Ki67^+^/nestin^+^ cells from 72 h to 7 days in SVZ [22]; Ki67 is a marker of cycling cells while nestin is an early marker of neuronal lineage cells. However, most of these proliferated cells could not survive more than 3 weeks due to extensive apoptosis [22]. 

Besides proliferation, the migration of DCX+ cells from SVZ to the perihematomal area has been observed by immunohistochemistry on days 14 and 28 after ICH induction [20]. Moreover, neuronal NeuN and BrdU double-positive cells are observed to be increased after 30 days in the hippocampal dentate gyrus of a subarachnoid hemorrhage (SAH) mouse model, which may be the result of proliferation, migration, and differentiation of NPCs [23].

Remyelination may be crucial for axonal maintenance and functional recovery due to the anatomical affinity of ICH, and deserves more attention. Hypertension-related ICH is the most common subtype of hemorrhagic stroke [24,25,26], often at the lenticulostriate artery and thalamus because of the rectangular structure [27]. The hematoma formed and enlarged here can compress adjacent white matter fibers, destroy myelin and axons, and leads to typical symptoms on the contralateral side. Prominent amounts of immunoreactivity for degraded myelin basic protein from degenerating myelin could be seen 1 day after collagenase injection, indicating acute and severe demyelination [28]. White matter tract injury shown on diffusion tensor imaging of ICH patients also supports the existence of demyelination [29]. Although much of our knowledge of white matter remyelination comes from the prototypical demyelinating disease multiple sclerosis, which shows that remyelination can be robust after lesion formation [30], it is plausible that there are remyelinating processes after ICH due to the capacity of oligodendrocyte lineage cells to repopulate a lesion. Indeed, in the collagenase-induced ICH model, both mature oligodendrocytes and oligodendrocyte precursor cells (OPCs) are markedly increased as determined by elevated CC1^+^Olig2^+^, NG2^+^Olig2^+^ and Ki67^+^Olig2^+^ cells, especially in the first 7 days post-ICH [31]; Olig2 is a transcription factor that defines cells of the oligodendrocyte lineage, while CC1 and NG2 are markers of mature oligodendrocytes and OPCs, respectively. These results demonstrate the existence of oligodendrogenesis after experimental ICH. Moreover, the proliferated Olig2^+^ cells are preferentially gathered inside the white matter bundles [31]. However, this research did not reveal whether these newly formed oligodendrocytes complete the remyelination process, requiring further investigation.

The basic primary pathology of ICH is the rupture of arterial walls and the formation of a hematoma, causing severe disruption of perfusion, circulation, and blood–brain barrier (BBB) permeability [32,33]. Because the brain is highly dependent on a blood supply for glucose and oxygen, reformation of the microvasculature is vital for post-ICH brain repair. Spontaneous angiogenesis can be observed around the hematoma in rats after collagenase-induced ICH [34]. The von Willebrand factor positive endothelial cells are noted to proliferate from 4 days and peak between 7 and 14 days after ICH as determined by increased BrdU^+^ counting. The level of vascular endothelial growth factor (VEGF) is also upregulated alongside its receptors including Flt-1 and Flk-1 [34]. 

In summary, the CNS has some degree of regenerative ability after ICH in animal models (Figure 1). The literature is still sparse, and it remains to be documented that the regenerative events are productive and lead to functional recovery. Nonetheless, the observation of tissue regeneration in ICH is promising and suggest that the process could likely be promoted if there is more knowledge of these reparative processes and of their mechanisms.

## 3. Evidence of CNS Regeneration in Patients with ICH

Brain samples from ICH patients are precious specimens to evaluate regenerative process after the insult. In the perihematomal region, markers of neuronal lineage cells indicating neurogenesis can be found, including doublecortin, the stem-like marker musashi1, and TUJ-1 and TUC-4 that indicate growth cone signaling; some of the NPCs are also Ki67 positive indicating proliferation [35]. As these samples were collected during 1-3 days post-ICH from hematoma removal surgery, the results suggest that neurogenesis occurs very soon after ictal onset.

Interestingly, postmortem research indicates that NPCs detected by p-Histone H3_Ser-10_ positive staining in the dentate gyrus in ICH patients are significantly higher than those with ischemic stroke [36], for reasons that are not clarified. In samples from subarachnoid hemorrhage patients, neurogenesis and NPC proliferation are demonstrated by Musashi 2^+^, Ki67^+^ and double-positive cells in the frontal lobe, while none of these labeled cells are seen in the control group [37]. 

Overall, signs of regenerative processes are seen after ICH in patients, but the studies are rare. Many more datasets must be forthcoming.

## 4. Neuroinflammation may Mediate Regenerative Processes after ICH: Guides from Experimental ICH

While the involvement of neuroinflammation in mediating secondary injury after ICH is well documented and reviewed [8,38,39], components of the inflammatory response also appear to be crucial to the reparative response after ICH (Figure 1). This should not be surprising since a fundamental role of an immune response is to help with healing after an insult. Here, we summarize immune cell subsets and their involvement in regenerative processes after ICH in preclinical models.

### 4.1. Microglia

Microglia comprise 5–10% of the total cell population in the normal brain. They are an early responder to injury in the CNS [38,40]. In a postmortem study, activated microglia are evident from day 1 after onset and dramatically increased to ninefold on day 12 post-ICH compared to controls [41]. 

In preclinical research, activated microglia are observed within the perihematomal area by 1 h of collagenase-induced ICH and 4 h post-autologous blood injection of ICH [42,43]. As microglia and infiltrated blood monocyte-derived macrophages display many similar markers and morphology, we refer to them as microglia/macrophages (M/M) when they have not been differentiated from one another in quoted studies. Accumulation of M/M is observed early after ICH [38] where their manifestation of pro-inflammatory (“M1”-like) phenotype leads to the release of a series of inflammatory cytokines, chemokines, matrix metalloproteinases (MMPs), free radicals, and other molecules that exacerbate neuroinflammation and secondary brain injury [44,45,46]. Depletion of microglia by colony-stimulating factor receptor 1 (CSFR1) inhibitor PLX3397 results in a reduction in lesion volume in the early stage (1 or 3 days post-surgery) of both collagenase and autologous blood-induced ICH [47]. However, the M/M cluster gradually transforms to a regulatory (“M2”-like) phenotype that increases from about 3 days post-ICH, peaks and dominates at around 14 days; these clusters are thought to promote tissue repair function including phagocytosis, remyelination, neurogenesis, and angiogenesis in CNS pathologies [39,48,49]. The intracerebral injection of interleukin-4 (IL-4) promotes the conversion of M/M to a regulatory phenotype displaying arginase 1 and improves functional recovery in ICH model [50]. The administration of another regulatory cytokine IL-10 intraventricularly increases the number of “M2”-like M/M and aids hematoma clearance by enhanced phagocytosis after injury [51]. 

The transcription factor nuclear factor-erythroid 2 p45-related factor 2 (Nrf2), a master regulator of antioxidative defense, may play a critical role in the phagocytosis ability of microglia. Nrf2 knockout mice have impaired hematoma clearance while administration of Nrf2 activator enhances such functions after ICH [52]. The activation of cannabinoid receptor-2 boosts “M2”-like microglia polarization in the acute stage after experimental ICH, which alleviates brain injury and neurological deficits [53]. Infiltrating macrophages contribute to hematoma clearance and tissue repair, and depleting peripheral monocytes with clodronate liposomes results in a larger lesion and neurological impairment [54]. In research about applying hirudin to improve long-term outcome after experimental ICH, M/M were found shifted to regulatory phenotype, and depleting M/M by PLX 3397 weakened the protective effect [55]. Recent research demonstrated that CD47 blocking antibody facilitated hematoma and iron clearance by microglia/macrophage while clodronate liposome-induced M/M elimination caused exacerbated brain edema, neuronal death, and functional deficits [56].

There is substantial literature describing the roles of M/M in mediating or promoting the extent of injury after ICH. We refer the reader to these reviews [38,57,58]. In contrast, the involvement of M/M in facilitating repair processes in ICH appears limited by the few citations noted above. Nevertheless, it is not clear from present data whether regulatory M/M directly promote CNS regeneration, or just neuroprotection after ICH. Work in this field is in urgent need and has substantial promises. 

### 4.2. Astrocytes

Astrocytes also display controversial functions in brain repair of ICH [59]. On the one side, reactive astrocytes generated after injury can evolve to excessive scar formation and inhibit repair processes [60]. They may increase lesion volume by generating pro-inflammatory molecules [60,61] and they produce inhibitory molecules including chondroitin sulfate proteoglycans, ephrins, and semaphorins that interfere with remyelination and functional recovery after brain injury [60]. Conversely, astrocytes are engaged with the repair of white matter tracts through promoting axonal regeneration and remyelination in stroke mice [62]. The neurotrophic factors secreted by reactive astrocytes facilitate the survival and migration of OPCs and NPCs that assist with remyelination and neurogenesis [62]. Inhibition of astrocyte reactivity by fluorocitrate weakens neurovascular reconstruction and worsens functional recovery in a mouse model of ischemic stroke [63]. Some subtypes of astrocyte are also reported to facilitate BBB recovery and white matter repair in the animal models of MS, spinal cord injury, neurodegenerative diseases, and tissue culture [64,65,66], but the data related to astrocyte in the repair phase of experimental ICH remain few and further investigation is essential.

### 4.3. Leukocytes

Infiltrated neutrophils could be observed within several hours after ICH onset and they accumulate over the next few days in the perihematomal area [41,67]; they are prominent sources of potentially detrimental molecules such as MMP-9, ROS, and TNF-α [8]. In support, higher blood levels of neutrophils after ICH correlate with perihematomal edema and poor functional recovery in patients [68,69]. 

However, on the bench side, a recent study reports a protective role of infiltrated neutrophils by secretion of lactoferrin, which aids the clearance of ferric iron, reduces injury and promotes functional recovery; this beneficial polarization of neutrophils is triggered by IL-27 from activated microglia after ICH induction in rats [70].

T lymphocytes can be found within the perihematomal region 1–3 days after symptom attack with moderate accumulation in ICH patients [41,71]. A meta-analysis showed that a low neutrophil-lymphocyte ratio predicted a better outcome including less probability of major disability and short-term mortality [72], which might suggest that the infiltration of T cells was generally beneficial for post-hemorrhagic recovery. The protective effect may come from regulatory T cells (Tregs) that promote M/M polarization to a regulatory phenotype in ICH models while deletion of Tregs exacerbates the injury [73,74].

It is plausible that ICH-induced inflammatory responses can mediate both salutary and adverse effects on brain regeneration and functional recovery depending on different stages and cellular phenotypes. Interventions aimed at harnessing the benefits of neuroinflammation are valuable for further investigation, and they have a highly translational potential for promoting tissue repair and improving prognosis for ICH patients.

## 5. Molecular Mechanisms Affecting Post-ICH Tissue Regeneration

Many of the molecules implicated in exacerbating tissue injury post-ICH appear also to be important in the subsequent repair process (Figure 2). 

As a nexus of coagulating response, thrombin is activated acutely in ICH lesions and it is implicated in BBB disruption, brain edema, and neuronal death in the early phase of ICH [75,76]. However, thrombin shows the ability to elicit neurogenesis by increasing DCX^+^ cells in experimental ICH, and this beneficial effect is neutralized by thrombin inhibition [77]. Post-ICH angiogenesis could also be triggered by thrombin in autologous blood model; proliferation of endothelial cells is dramatically upregulated following thrombin injection as well as levels of VEGF, hypoxia-inducible factor-1α (HIF-1α), and angiopoietin-1 (Ang-1) and Ang-2, which are all crucial factors for angiogenesis [78]. These thrombin-induced regenerative processes might be attributed to the activation of protease-activated receptor-1 (PAR-1) [79,80,81]. Moreover, pericyte (an essential component of a neurovascular unit) coverage is significantly reduced by thrombin inhibition in ICH rats, which indicates the support of BBB reconstruction by thrombin [82]. In addition, thrombin facilitates the synthesis and secretion of nerve growth factor in glia which improves neurite outgrowth after brain injury [83].

The role of MMP-9 has been widely explored as an injurious factor in ICH. It can degrade the extracellular matrix [84,85,86] and contributes to BBB disruption; it enhances inflammation, perihematomal edema, and worse lesional outcome after ICH [87,88,89,90,91,92], especially in the acute phase. However, more recent data show that increased MMP-9 may promote neurogenesis and angiogenesis in an ICH model, whereas intraventricular administration of MMP-9 siRNA on day 7 and 10 reduces the level of VEGF and NGF as well as DCX and BrdU positive cell counts at day 10 and 14 post-injury [93]. It is proposed that MMP-9 displays dual effects in ICH, and the key to utilizing its advantage might be its administration at particular points after stroke [94].

High mobility group box1 (HMGB1) is a highly conserved DNA-binding protein related to inflammatory responses by triggering various pattern recognition receptors (PRRs) after ICH and has both beneficial and harmful effects [95,96,97,98,99]. Studies focused on regeneration found that HMGB1 level is positively correlated with VEGF, BDNF, and NGF concentration after collagenase induced ICH; inhibition of HMGB1 reduces the number of proliferating cells and neural precursor cells as well as the level of trophic factors, supporting the promotive function of HMGB1 on neurogenesis and angiogenesis [100,101]. Further investigation reveals that the neurogenic and angiogenic effect may be due to the activation of the receptor for advanced glycation end-products (RAGE), a pattern recognition receptor, by HMGB1 in the later phase after experimental ICH [101,102]. Toll-like receptor 4 (TLR4) is also a member of pattern recognition receptors, expressed on neurons and glia, which acts in diverse roles in CNS injury [103]. In experimental ICH, TLR4 stimulation aggravates DNA damage, neuronal death, and neurological deficits at 1, 3, and 5 days after injury [104]. However at 14 days post-ICH, the TLR4 antagonist TAK-242 lowers NPC counts and content of BDNF and VEGF, suggesting that delayed TLR4 activation may improve brain tissue recovery [105].

Lactate accumulation is observed in patients and in a pig model of ICH [106,107]. This metabolic intermediate of glycolysis was once regarded as a symbol of neuronal dysfunction until lactate was found to assist synaptic activity and be utilized by neurons [108,109,110]. Inhibition of lactate dehydrogenase markedly reduces the number of newly formed endothelial cells and NPCs, whereas exogenous lactate administration boosts angiogenesis and neurogenesis by increasing NFκB translocation [111].

The Human Tp53 gene regulates the apoptotic activity of tumor suppressor protein P53 in physiological conditions, displaying a single-nucleotide polymorphism (SNP) at codon 72 with an arginine-to-proline amino-acidic substitution [112,113,114,115]. Recent data show that patients harboring the proline allele have better functional outcomes with a higher VEGF level and circulating CD34^+^ endothelial progenitor cells than homozygous arginine patients after ICH [116], which may occur due to better neurovascularization [117]. Humanized proline allele knock-in reduces apoptosis of CD31+ endothelial cells in collagenase-induced ICH mice while promoting angiogenesis and functional recovery compared to the arginine allele carrying group [116]. 

Several other mediators with limited data are also implicated in repair after ICH. STAT3 regulates many important physiological functions including cellular proliferation, migration, and angiogenic activity [118]. However, STAT3 signaling may interfere with neurogenesis and recovery after experimental ICH and inhibition of the pathway increases proliferating neural progenitor cells around the hematoma [119]. Nogo-A is a myelin-related protein exerting an inhibitory effect on axonal growth after brain injury [120,121]. ICH-induced activation of Nogo-A and its downstream signaling pathway impair neuronal survival and axonal regeneration in a rat model of collagenase injection [122]. Notch1 signaling is related to neurogenesis in the adult brain, which regulates proliferation and differentiation of neural stem cells [123]. In the autologous blood ICH model of mice, upregulating Notch1 expression increases the number of neural stem cells in the hippocampal dentate gyrus while promoting performance in behavior tests on day 28 after injury [124].

The research about molecular mechanisms related to regeneration after ICH remains sparse. It would be an urgent task to uncover more data and integrate them to acquire a broad interacting molecular network of repair processes after ICH.

## 6. Strategies to Promote Neural Regeneration after ICH

### 6.1. Medications

Many medications or substances have been observed to confer neuroprotection and lead to improve functional recovery or histological outcomes in animal models, and we refer the reader to excellent reviews elsewhere [8,125,126]. Here we focus on the drugs that show potential on promoting tissue repair in preclinical research of ICH.

Statins have been widely used to prevent ischemic stroke [127,128], but concerns about increasing the risk of ICH remain [129,130]. However, growing evidence suggests that statin use in primary or secondary prevention of ischemic stroke does not elevate the risk of acquiring ICH [131,132,133]. The ongoing SATURN trial may resolve the safety of statins in ICH patients. Interestingly, a pilot study reported lower mortality of ICH patients in the rosuvastatin treatment group [134]. Additional data from a cohort study lasting 10 years in Denmark report that stroke-free statin users had a 22–35% lower risk for ICH compared to reference subjects [133]. In preclinical ICH, the results of statin treatment are usually beneficial. In an autologous blood induced model, simvastatin or atorvastatin given to rats daily for 1 week post-injury augments the number of DCX^+^ neural precursor cells and BrdU^+^ proliferating cells along with better neurological functions compared to controls at 28 days; ameliorated tissue loss at this time point is observed by both MRI and histology [135]. In the same model, enhanced neurogenesis and synaptogenesis is reported by an earlier study that shows increased DCX, synaptophysin and TUJ 1 positive cells with treatment of statin [136]. Statin is also documented to stimulate the generation of VEGF, BDNF, and NGF, which may facilitate repair after ICH through neurotrophic ways [137]. Angiogenesis and revascularization can be promoted by statin in rat model, observed in both histology and MRI [138]. In another study, statin was found to stimulate “M2”-like polarization of microglia with enhanced phagocytosis function, which promotes hematoma and iron clearance, leading to better tissue and functional recovery [139]. Moreover, statin may act as a immunomodulator to inhibit excessive inflammatory response in the early phase after experimental ICH to limit secondary brain injury [8]. 

Minocycline is often tested as a microglia inhibitor to control injurious neuroinflammation soon after experimental ICH where it displays various therapeutic effects [140,141,142]. The inhibition of proinflammatory microglia phenotype seemingly does not interfere with the regulatory properties of microglia [143]. In autologous blood induced ICH rats, administration of minocycline facilitates an “M2”-like polarization and increases microglia-derived BDNF; enhanced neurogenesis is observed with more DCX and Tuj-1 positive neuron-like cells than in a control group at 24 h after ICH onset [144]. Another study reported NGF elevation by minocycline after collagenase ICH [145]. However, minocycline injection is documented to inhibit angiogenesis by downregulating the level of VEGF and its receptors after experimental ICH, which may hinder tissue regeneration in the late phase [146]. For now, all completed clinical trials demonstrate safety but not efficacy of minocycline treatment for cerebral hemorrhage [147,148], although the studies are not powered for efficacy. The roles and usage of minocycline in ICH still need more investigation from both clinical and preclinical work.

Four of five sphingosine-1-phosphate receptors (S1PR1, S1PR2, S1PR3, and S1PR5) are found expressed at different levels in divergent neural cells of the CNS. S1PR1 activation is reported to be linked to neuronal growth, reduced proinflammatory microglial activity, and myelin formation, while S1PR5 signaling assists mature oligodendrocyte survival [149]. A multiple S1PRs modulator, fingolimod, is shown to elevate neurotrophic factors including BDNF and GDNF in cultured microglia, and to promote regulatory polarization while inhibiting the proinflammatory property of microglia in an ischemic model [150,151]. In mice with collagenase induced ICH, 4 weeks of fingolimod treatment post-injury significantly improves white matter integrity, neuronal survival, and functional performance at 28 days without altering lesion volume at 5 days [152], which implies that fingolimod facilitates tissue repair in the later phase. It would be reasonable to hypothesize that S1PRs modulators may contribute to regenerative processes including neurogenesis and remyelination after ICH, but direct evidence is lacking. Although many data show that S1PR modulators improve functional recovery in different ICH models, the studies have focused on inhibition of harmful neuroinflammation [39,153,154,155] that can lead to secondary improvement. A proof-of-concept clinical study documented reduced perihematomal edema and better functional recovery after oral administration of fingolimod for three days post-onset comparing to patients with standard care [156]. 

Siponimod is a selective S1PR modulator predominantly binding to S1PR1 and S1PR5 [149], which may generate more protective effects and a less adverse reaction in ICH. The therapeutic effects of siponimod have been demonstrated in both collagenase and autologous blood models of ICH, but these studies did not address tissue repair [144,157,158]. A phase II randomized, placebo-controlled, double-blind clinical trial of siponimod in ICH patients is ongoing but temporarily suspended due to COVID-19.

Lithium, a mood stabilizer for bipolar disorders [159], was recently shown to be beneficial in preclinical ICH. Intraperitoneal administration of lithium chloride immediately after ICH induction promotes the “M2”-like polarization of microglia with enhanced phagocytosis and hematoma resolution within first 7 days post-injury; elevated levels of VEGF and BDNF that may contribute to angiogenesis and neurogenesis are documented in the subsequent 7 days [160]. Lithium has also been found to alleviate white matter injury including demyelination, axonal degeneration, and death of oligodendrocytes in the autologous blood ICH model, correspondent with upregulated BDNF level [161]; it is uncertain whether lithium chloride promotes white matter repair or protects from injury. 

CD47, an integrin-associated protein expressed on erythrocytes, has been demonstrated to regulate hematoma clearance in the ICH model [162]. A blocking antibody to CD47 improves behavioral performance while reducing lesion volume by boosting M/M-induced erythrophagocytosis after experimental ICH [56,163]. 

Neurotrophins are essential and beneficial for tissue repair after brain injury, but exogenous neurotrophic factors are hard to sustain at a therapeutic concentration in the lesion area; chemical modifications may resolve this problem. Brain-derived neurotrophic factor (BDNF) fused to a collagen-binding domain could stimulate neurogenesis and angiogenesis better than natural BDNF and it maintains the growth factor at a higher level in the injured hemisphere after injection into the lateral ventricle of rats with ICH [164]. Exogenous fibrin-binding domain fused BDNF is observed to concentrate in the perihematomal area and to promote neural regeneration with ameliorated neurological deficits [165]. Moreover, there is a proof-of-concept study that administration of mouse nerve growth factor improves 3 month functional recovery compared to citicoline controls in patients with spontaneous ICH [166]. 

Besides pharmaceutical methods, many other strategies with potential to promote tissue repair and functional recovery after ICH in preclinical studies or clinical trials have been proposed (Table 1). Here we summarize these promising non-medication treatments and their possible mechanisms in promoting tissue repair after hemorrhagic stroke to provide information for further research and translation.

### 6.2. Stem Cell Therapy

As a promising strategy to improve the dismal prognosis, stem cells and related therapies remain popular in the realm of ICH research, and they have safety and improved functional outcomes in several clinical trials [167,168,169,170]. The therapeutic effects of stem cells could mainly be attributed to cell replacement proliferation and differentiation to neurons or glial cells, and/or the paracrine secretion of multiple neurotrophins and regulatory molecules [171,172,173] to assist immunoregulation, neural cell survival, and tissue repair, after CNS injury [174,175,176,177]. In preclinical studies, the administration of different types of stem cells or their products is observed to promote neural regeneration and recovery.

Mesenchymal stem cells (MSCs) are the most widely used cell type in research of ICH treatment [178]; they reduce lesion volume and inflammation while increasing angiogenesis, tissue repair, and functional recovery in different animal models of ICH [179]. Bone-marrow-derived mesenchymal stem cell (BMSC) transplants proliferate and differentiate into neural cells and increases the level of BDNF after collagenase induced ICH [180]. Axonal sprouting and regeneration, and improved functional recovery, are enhanced by transplantation in the same model [181]. In the hemoglobin-induced ICH model, BMSC grafts increase NeuN^+^ (marker of mature neuron) cells and upregulate ZO-1 (a part of tight junction) expression as well as decreasing inflammatory response [182]. BM-MSCs are also observed to promote axonal regeneration in the autologous blood ICH model, which might be mediated by activating ERK1/2 and PI3K/Akt signaling pathways [183]. 

Some researchers have tried to facilitate the therapeutic effects of stem cells by genetic manipulation. Glial cell line-derived neurotrophic factor (GDNF) plays a crucial role in differentiation, survival, and repair in CNS [184,185,186]. GDNF transfected MSCs express neural cell-specific biomarkers including NSE, MAP2, and GFAP after implantation into ICH rats which leads to better behavioral performance than parental MSCs [187]. Moreover, overexpression of microRNA-126a-3p in BM-MSCs appears to repair the blood–brain barrier by differentiating to CD31^+^ endothelial cells and upregulating ZO-1 and claudin-5 (both tight junction proteins) after ICH in rats [188]. 

Multi-lineage differentiating stress enduring (Muse) cell, a novel type of non-tumorigenic pluripotent stem cell, shows high potential for tissue regeneration and lesional navigation, and can be collected from cultured mesenchymal stem cells by stage-specific embryonic antigen 3 and CD105 double-positive sorting [189,190,191]. In the autologous blood-induced ICH model of mice, human Muse cells administrated into the hematoma cavity 5 days after injury survived well and led to better functional recovery than MSC controls at day 69, associated with significantly higher ratio of NeuN (57%) and MAP2 (41.6%) [192]. 

Adipose-derived stem cell (ADSC) is also a member of mesenchymal stem cells. Intravenous injection of human ADSCs in the acute phase after experimental ICH results in alleviated neurological deficits during the subacute phase [193]. Cerebral ventricle administrated ADSCs are observed to differentiate into neuron-like and astrocyte-like cells and upregulate VEGF level as well as promoting neurological functions [194]. Brain edema and tissue damage are reduced by ADSCs implantation in another ICH model [195]. CX3CR1 is a receptor of the chemokine fractalkine [196,197,198]. Overexpression of CX3CR1 of ADSCs facilitates the migration ability of engrafted cells to the perihematomal region of ICH mice and improves scores in behavioral tests compared to naïve stem cells [199]. Umbilical tissue can be another resource for mesenchymal stem cells. Intravenous administration of umbilical cord-derived stem cells (UCSC) improves neurogenesis (marked by BrdU, TUJ1, DCX, NeuN and synaptophysin) and angiogenesis (marked by vWF) as well as motor functions after autologous blood injection into rat striatum [200]. In addition, intraventricular engraftment of hepatocyte growth factor transfected UCSCs 1 week after collagenase-ICH promotes tissue repair and neurological recovery by remyelination and axonal regeneration [201]. 

Neural stem cells (NSCs) and induced pluripotent stem cells (iPSCs) were commonly studied in ICH treatment about 10–20 years ago [178]. A result published in 2003 showed that intravenous administration of human NSCs one day after collagenase-induced ICH in rodents exhibited better functional performance with injected cells migrating to the injury region where 10% differentiated to neuron-like cells, whereas 75% became glia [202]. Applying immortalized human NSCs and intracerebral delivery seems to increase neural-like differentiation (30–40%) of engrafted cells [203]. Moreover, intracerebral transplantation of fetal neural stem cells or cell-conditioned medium both improve neurological function [204]. Intracerebral implantation of iPSCs promotes functional recovery and reduces neuronal death after experimental ICH, but engrafted cells predominantly differentiate to GFAP positive astrocytes [205]; conversely, human iPSCs derived neuroepithelial-like stem cells mature and transform into neurons in the post-ICH microenvironment [206]. 

Disadvantages of iPSCs include high tumorigenic risks [207], and NSCs may be harder to prepare and proliferate than other stem cell populations, which could be the reason why they are not that popular in ICH research today. Recently, stem cell-derived exosomes, the main component of therapeutic paracrine mechanisms, which contain proteins, RNAs, and lipids that might mediate tissue repair and immunomodulation after CNS injury, are attracting more interest [208,209]. In preclinical studies of ICH, intravenous administration of MSC-derived exosomes dramatically promotes white matter repair, axonal sprouting, and functional restoration [210]. Another study reports that injection of proteins from MSC-derived exosomes increased myelin coverage and endothelial cells in the perihematomal area as well as neuroblasts and mature neurons in the subventricular zone after blood injection induced ICH; both cognitive and sensorimotor function are restored by the treatment [211]. Furthermore, exosomes derived from genetic modified MSCs also promote functional recovery and neural survival [212,213].

Exosomes are easier to prepare and store in large amounts and have a lower potential for tumorgenicity, immunogenicity, and thrombosis than stem cells, making them more feasible in clinical application. However, their therapeutic effects might be restricted because exosomes cannot play a part in direct tissue replacement that may be crucial in regeneration after ICH. In general, stem cells and their products are highly promising for translation and improve the dismal prognosis of ICH, but challenges include safety, type, timepoint, dosage, and mode of delivery.

### 6.3. Biomaterials and Nanoparticles

The interdisciplinary cooperation between material science and medicine has become common, including for ICH. Hydrogel is considered a biocompatible material that can be injected during minimally invasive surgery and form a matrix for cell infiltration and adhesion to facilitate tissue repair after stroke [214,215]. Gelatin hydrogel injection into the lesion three days post collagenase-induced ICH is reported to alleviate neurological deficits of mice due to conversion of M/M from pro-inflammatory to regulatory [216]. Moreover, hydrogels may carry medications, neurotrophic factors, and stem cells as well as receive chemical modifications to enhance therapeutic effects [215,217]. Hydrogel containing epidermal growth factor (EGF) significantly increases the number of neural precursor cells (nestin-positive) around the lesion of ICH rats compared to hydrogel or EGF alone, with some of the cells differentiating to TUJ1^+^ neurons; neurological recovery is better in the EGF-hydrogel group [218]. Self-assembling peptide nanofiber scaffolds (SAPNS) can eventually become hydrogels after delivery and promote wound healing [219]. 

RADA16-I is a type of SAPNS that after injection combined with hematoma aspiration improves functional recovery in experimental ICH but almost no neurons or nerve fibers were found in the matrix [220]; a modification was made to alter its acid property to neutral, which led to nerve fibers growing within and better behavioral performance [221]. Intravenous administration of ceria nanoparticles aids OPCs proliferation, maturation, and remyelination in the collagenase model; EdU^+^CC1^+^ (proliferated mature OLs) and Oligo2^+^CC1^+^ (mature OL lineage cell) cell counting is elevated at 7 days after injury compared to vehicle treatment, while MBP positive area and thickness of myelin sheath are increased at 21 days [222]. 

### 6.4. Rehabilitation Training

Rehabilitation has been widely used for decades to improve patients’ functional recovery after ICH and shown stable efficacy [223,224]. Growing evidence from clinical trials and a cohort study suggest that rehabilitation should be implemented as early as possible and continue for a more extended period [225,226,227,228]. However, the method of post-ICH rehabilitation has followed the same pattern of ischemic stroke in clinical practice. The data from preclinical studies may give us inspiration for ICH specific rehabilitation. Skilled reach training requires animals to reach food through a narrow gap with one forelimb, which promotes astrocyte process growth, dendritic reorganization, and BDNF level, as well as improving sensorimotor functions in the collagenase induced ICH model [229,230]. An enriched environment contains tunnels, toys, and others to provide animals with multiple forms of sensory stimulation and more opportunities for physical activity, which not only improves functional recovery but also increases the dendritic length and reduces lesion volume and neuronal death after experimental ICH when combined with skilled reach training [231,232,233,234]. Acrobatic training provides a route that includes various types of barriers for mice to walk through repeatedly; motor function and coordinated movement ability are significantly restored after training in the collagenase model, and enhanced neuronal activity and synaptic remodeling are also observed [235]. Application of treadmill running, a type of aerobic training on post-ICH animals, induces longer dendritic length, complexity, and lower motor deficits [236]. Although some results suggest that rehabilitation may reduce lesion or cell death, most of the mechanisms elucidated in preclinical studies are related to neuronal or synaptic plasticity, partially attributed to astroglial activity. Recently, some innovative techniques of rehabilitation have emerged in clinical studies of ICH. Vagus nerve stimulation added to rehabilitative training prompts a much higher rate of functional recovery than patients of the rehabilitation-only group [237]. A clinical trial for robot-assisted therapy displays ameliorated neurological deficits of stroke (ICH included) patients compared with the non-physical trained group but not with patients accepted for intensive rehabilitation [238]. Nevertheless, more beneficial attempts are encouraged and necessary to alleviate disability after ICH by exploring advanced technology and task designs that may be distinctive for hemorrhagic stroke due to different anatomical predilection and pathophysiology from ischemic stroke.

## 7. Conclusions and the Future

The advances of minimally invasive surgery have contributed meaningfully to ameliorate the poor outcome of patients with supratentorial cerebral hemorrhage [239]. Moreover, immunotherapy targeting neuroinflammation seems very promising to alleviate ICH-induced secondary brain injury. Despite this, most of the interventions have displayed safety without efficacy according to available clinical trial results [8]. Moreover, hemorrhagic stroke shows a significantly younger trend in developing countries, and the prognosis remains very undesirable [240,241]. More attention must be devoted to research on regaining functions such as through neural cell repopulation and axonal remyelination. There is good evidence for brain repair after human ICH, so it is realistic to promote this reparative process. However, research on the regenerative process in ICH patients remains rare and approaches have been restricted to the short period after onset and have only focused on neurogenesis. It would be imperative to extend comprehensive regenerative studies to the later periods post-ICH. Although several unknowns remain, what we do know is that neuroinflammation, particularly microglial activity, is intimately related to post-ICH restoration, so eliciting the regulatory/reparative phenotype of these cells could be a target for manipulation. With respect to fostering regeneration, drugs and bioactive substances may be easier for translation but the precise dosage, usage, and timepoint of drug treatment must be investigated to optimize its therapeutic effects during the reparative stage of ICH. Stem cell therapies are powerful tools on promoting recovery as they can repair injured tissue directly by cell replacement and/or facilitate endogenous repair by paracrine mechanisms. However, it would take some time before high-quality large multi-center clinical trials of stem cell treatment in ICH can occur due to concerns about safety and ethics, while the difficulty of preparation of cells is another barrier.

Since intracerebral hemorrhage is a complex intractable disease with intricate pathology, an integrated strategy for treatment deserves consideration. It would be worthwhile to imagine that the disastrous outcomes of ICH can be reversed in the future when stem cells are implanted into the cavity of a patient’s brain after minimally invasive surgery followed by potent pro-regenerative drugs and immunomodulators, as well as choreographed rehabilitation tasks.

## Figures and Tables

**Figure 1 cells-10-02513-f001:**
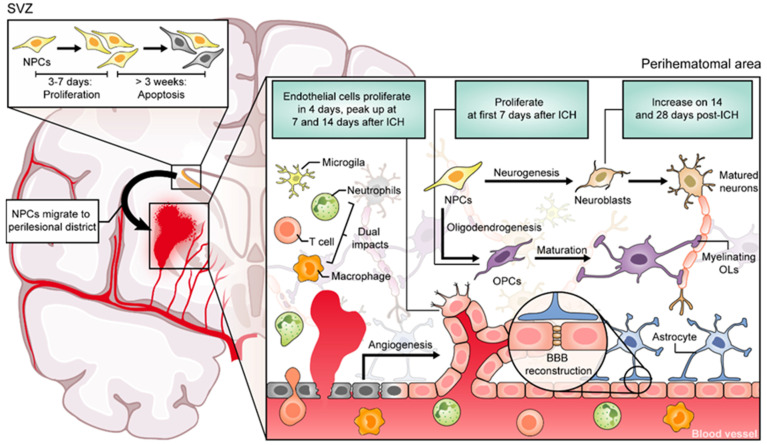
Signals from tissue damage and inflammatory cells may trigger the process of repair. Neural precursor cells (NPCs) proliferate from SVZ and migrate to the perihematomal area and differentiate into neurons and glia, contributing to brain restoration. However, most of these NPCs could not survive more than 3 weeks due to extensive apoptosis. Moreover, the proliferated Olig2^+^ cells were preferentially gathered inside the white matter bundles, although there are no data on whether these newly formed oligodendrocytes contribute to subsequent remyelination. Inflammatory products from resident and infiltrated immune cells promote or hinder tissue recovery according to their phenotypes.

**Figure 2 cells-10-02513-f002:**
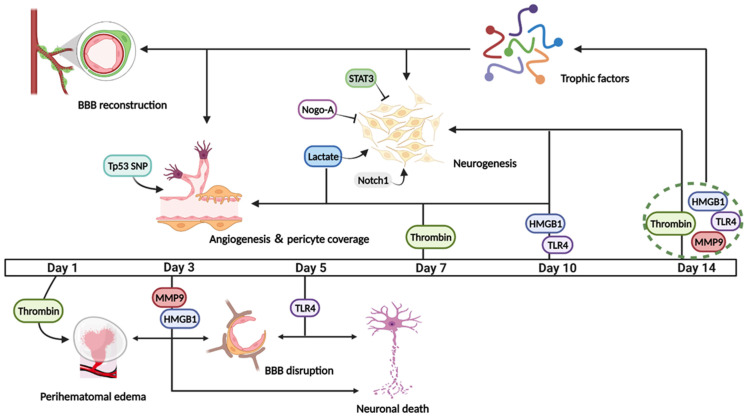
Some molecules including thrombin, MMP-9, HMGB1, TLR4, show divergent effects on post-ICH repair. They tend towards causing pathological changes such as brain edema, BBB disruption, neuronal degeneration, or death in the early phase after ICH, but convert to beneficial regenerative processes by increasing DCX+ cells, promoting endothelial proliferation, pericyte coverage, and stimulating trophic factors at the later stage after ICH. Moreover, the expression of STAT3 and Nogo-A after experimental ICH is observed to inhibit neurogenesis, while Notch1 and the generation of lactate can promote this process.

**Table 1 cells-10-02513-t001:** Potential strategies to promote tissue regeneration after intracerebral hemorrhage.

Approaches		Possible Benefits
Medications	Statins	Neurogenesis, angiogenesis, phagocytosis
	Minocycline	“M2” polarization, neurogenesis
	Fingolimod/Siponimod	Neurogenesis, remyelination
	Lithium	Trophic factors, white matter repair
	CD47 antibody	Hematoma clearance
Stem cells	BM-MSCs	Axonal regeneration, BBB reconstruction
	Muse cells	Cell replacement of neurons
	ADSCs	Neuron-like differentiation, VEGF
	UCSCs	Neurogenesis, angiogenesis, remyelination
	Exosomes	Remyelination, axonal sprouting, neurogenesis
Biomaterials	Hydrogel	Regulatory polarization, neurogenesis
	SAPNS	Oligodendrogenesis, remyelination
Rehabilitation	Skilled reach training	Dendritic reorganization, BDNF
	Enriched environment	Dendritic length
	Acrobatic training	Neuronal activity, synaptic remodeling
	Aerobic training	Dendritic length and complexity

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
