# Peer review of "Central Nervous System Tissue Regeneration after Intracerebral Hemorrhage: The Next Frontier"

_cells, 2021, doi:10.3390/cells10102513_

Round 1
Reviewer 1 Report
Zhang et al. review the literature on the process and mechanisms of tissue restoration/regeneration after ICH. The literature review is well written and thorough well done. A "Table" summarizes the strategies to promote neural regeneration after ICH (including medications & cell therapy) will strengthen this review article.
Author Response
Thank you for the comment that the review is well written and thorough. We have now included a table that summarizes the strategies to promote neural regeneration after ICH. It is currently placed under line 433 in the text.
We agree that the table improves the readability of the review, as one can refer to that quickly.
Reviewer 2 Report
The reviewer is neurosurgeon with intimate relationship to the ICH, especially how to minimize primary trauma after stroke via minimally invasive approaches and he is preferring urgent surgery, because big part of hematomas show tendency to growth in volume. Stroke protocol is widely spread around the world and this fact allows early treatment also for hemorrhagic types of stroke.
But how to minimize secondary trauma of brain tissue and how to support effective regeneration of brain tissue is out of neurosurgical possibilities.
Authors review brings nice overview about state of art in this field and could be helpful for people interesting for this topic. Socially and economically independent patient after ICH is still the run to the long distance, but this review could be for somebody first step in this field.
Author Response
Thank you for the favourable comment that this review brings a good overview about the state of the art in the field of repair in intracerebral hemorrhage.
We have addressed the concern of Reviewer 1 and have added a table that now improves readability.
We appreciate your endorsement.